# Biocontrol of *Fusarium solani*: Antifungal Activity of Chitosan and Induction of Defence Enzymes

**DOI:** 10.3390/plants14030431

**Published:** 2025-02-01

**Authors:** Juan Antonio Torres-Rodriguez, Juan José Reyes-Pérez, Mercedes Susana Carranza-Patiño, Robinson J. Herrera-Feijoo, Pablo Preciado-Rangel, Luis Guillermo Hernandez-Montiel

**Affiliations:** 1Facultad de Ciencias Agrarias y Forestales, Universidad Técnica Estatal de Quevedo, Av. Quito km 1.5 vía a Santo Domingo, Quevedo 120501, Ecuador; jreyes@uteq.edu.ec (J.J.R.-P.); mcarranza@uteq.edu.ec (M.S.C.-P.); rherreraf2@uteq.edu.ec (R.J.H.-F.); 2Tecnológico Nacional de México/Campus Instituto Tecnológico de Torreón, Carretera Torreón-San Pedro km 7.5, ejido Ana, Torreón, Coahuila 27170, Mexico; pablo.pr@torreon.tecnm.mx; 3Nanotechnology & Microbial Biocontrol Group, Centro de Investigaciones Biológicas del Noroeste, La Paz, Baja California Sur 23096, Mexico

**Keywords:** growth inhibition, tomato plants, pathogen resistance, cellular damage, antioxidant enzymes

## Abstract

In this work, the efficiency of chitosan as a biocontrol agent against *Fusarium solani* on tomato plants was determined and the antifungal activity and the induction of defence enzymes were evaluated. Treatments were carried out with different concentrations of chitosan (1, 2 and 3 g L^−1^) combined with a synthetic fungicide (carbendazim). The results showed that all chitosan treatments significantly inhibited the mycelial growth and biomass of *F. solani*, with the most effective results obtained with the 3 g L^−1^ treatment. Scanning electron microscopy revealed that chitosan causes severe structural damage to *F. solani*, including cell lysis and the deformation of mycelium and spores. In addition, plants treated with chitosan showed significant improvements in height, stem diameter, root dry biomass and root length compared to those treated with synthetic fungicide and the control (no chitosan application). Enzyme assays showed that chitosan significantly increased superoxide dismutase, catalase, peroxidase and phenylalanine ammonia-lyase activity, indicating an increased defensive response. These results suggest that chitosan is a viable and less toxic alternative for the management of disease caused by *F. solani* in tomato plants, promoting both plant health and growth.

## 1. Introduction

Tomato (*Solanum lycopersicum* L.) is a vegetable of significant economic and nutritional importance. Economically, tomatoes are the second most important vegetable crop in the world, with production exceeding 180 million tonnes per year [1]. It is central to the global agricultural economy due to its high demand in both fresh and processed markets [2]. However, tomato is susceptible to root rot caused by the fungus *Fusarium solani* [3]. Damage caused by this phytopathogen includes root and stem rot, plant wilting, vascular discolouration and, in severe cases, plant death [4,5].

Seventy percent of plant diseases are caused by fungi and similar organisms, which pose a threat to agriculture and food security [6]. Even with the advent of new fungicides, these fungi have a high adaptive potential to infect plants [7]. In addition, *Fusarium* species have the ability to remain in the soil for decades, even without the presence of a host plant, making them difficult to control [8].

Synthetic fungicides, although effective in controlling plant diseases, have several harmful effects on both the environment and human health [9,10]. In addition, some synthetic fungicides have been found to have negative effects on non-target microorganisms, reducing microbial diversity and affecting beneficial species [11,12].

Fungicides used for the control of *F. solani* include carbendazim, a broad-spectrum systemic fungicide belonging to the benzimidazole class. Its mechanism of action is based on the inhibition of microtubule formation during fungal cell division, interfering with the ability of the fungus to grow and reproduce [13,14].

In the current search for sustainable alternatives for the biocontrol of phytopathogens, chitosan has emerged as a promising solution due to its biological and ecological properties [15,16]. Chitosan, a polymer derived from the deacetylation of chitin, is mainly obtained from the exoskeletons of crustaceans, such as shrimps and crabs, as well as certain insects and fungi [17]. Chitin was the first polysaccharide identified by humans, discovered some 30 years before cellulose, and is the most abundant polymer after cellulose [15]. Chitosan bi-polymer is biodegradable, biocompatible and non-toxic, making it an option for applications in agriculture, medicine and other industrial fields [16].

Applications of chitosan in agriculture include the protection and stimulation of seed germination through coatings and induction of plant growth and development [18,19]. It has also been shown to increase plant growth and yield [20,21] by influencing several physiological processes, such as nutrient uptake, cell division and elongation, reduced transpiration, enzyme activation and protein synthesis [22]. In addition, chitosan can mitigate the negative effects of abiotic stress, improve soil properties, prevent nutrient leaching and chelate cationic and anionic heavy metals [23,24,25].

The antifungal activity of chitosan is mainly due to its ability to disrupt the integrity of the fungal cell membrane, interfering with its energy metabolism and electron transfer chain [26,27]. This disruptive action occurs because chitosan, being a cationic polymer, interacts with microorganisms’ negatively charged cell surface and destabilises the cell wall [16,28]. Another mechanism involves chitosan’s interaction with microbial DNA, inhibiting mRNA transcription and affecting protein synthesis [29,30]. In addition, chitosan exhibits chelating properties, allowing it to chelate metal ions and essential nutrients that are necessary for microbial cell growth, affecting their availability and, consequently, inhibiting microbial growth [31,32].

Moreover, chitosan acts as an inducer of defence responses in plants. Its application increases the production of phytoalexins, defence-related enzymes, pathogen-related proteins (PRs) and lignin, as well as inducing callose formation, thus strengthening the natural defences of plants against phytopathogens [16,33].

The aim of this research was to determine the effect of chitosan for the biological control of *Fusarium solani* in tomato crops. This study aims to provide solid data to support the use of chitosan as a biotechnological tool that not only contributes to the protection of tomato crops but also promotes more sustainable and environmentally friendly agricultural practices.

## 2. Results

### 2.1. Antifungal Activity of Chitosan 

All chitosan treatments inhibited the mycelial growth of *F. solani*. An increasing chitosan concentration was related to the highest inhibition of the phytopathogen. The highest inhibition of *F. solani* was obtained with the concentration of 3 g L^−1^ with a mean of 81.25%, significantly higher (*p* < 0.05) than the other chitosan treatments and the synthetic fungicide. Treatments 2 and 1 g L^−1^, with means of 76.14 and 65.15%, respectively, were less effective in inhibiting *F. solani* compared to the synthetic fungicide (Figure 1).

### 2.2. Chitosan’s Antifungal Activity on Fusarium solani Biomass

The chitosan treatments showed significant differences (*p* < 0.05) in the inhibition of *F. solani* biomass with respect to the control treatment. The synthetic fungicide treatment (carbendazim) showed the highest biomass inhibition, with an average of 0.65 g. The best chitosan treatment was 3 g L^−1^, showing significant differences (*p* < 0.05) with the 2 and 1 g L^−1^ treatments. The treatment with the lowest chitosan concentration (1 g L^−1^) showed a mean biomass of 2.09 g, making it the least effective chitosan treatment (Figure 2).

### 2.3. Effect of Chitosan on the Cell Structure of Fusarium solani

Figure 3 shows the effect of chitosan on the cell structure of *F. solani* using scanning electron microscopy (SEM). All chitosan treatments damaged the cell structure of *F. solani* compared to the control treatment. The control treatment showed a dense and uniform structure with a large number of well-formed and clustered spores. The damage in the structure was evident in the hyphae of the cells treated with the chitosan treatment (1, 2 and 3 g L^−1^) and it was observed that they presented cell lysis. In addition, the mycelium and spores of *F. solani* were deformed, irregular and reduced in number. The greatest damage to the cellular structure of *F. solani* was observed with the 3 g L^−1^ treatment, where the spores (3 g L^−1^ D) and mycelium (3 g L^−1^ D) were completely deformed compared to the control (non-infested plants) treatment (Control A), which did not show any damage to the cellular structure.

### 2.4. Chitosan Biocontrol Against Fusarium solani on Tomato Plants

The 3 g L^−1^ treatment presented a plant height of 330.42 mm, showing significant differences (*p* < 0.05) with the rest of the chitosan treatments and the synthetic fungicide. The 2 g L^−1^ chitosan treatment also showed a significant increase in height (321.67 mm), outperforming the carbendazim treatment (309.82 mm) and the 1 g L^−1^ chitosan treatment (293.73 mm). The *F. solani* treatment showed the lowest height (155.90 mm), evidencing the severe damage caused by infection without the use of chitosan or a synthetic fungicide.

The chitosan treatments of 3 g L^−1^ (4.29 mm), 2 g L^−1^ (3.7 mm), and carbendazim (3.62 mm) did not present significant differences (*p* > 0.05) in stem diameter. However, they showed significant differences (*p* < 0.05) with the *F. solani* treatment (2.32 mm), which had the smallest stem diameter.

The highest root dry biomass was obtained with the chitosan treatment at 3 g L^−1^ (68.33 mg). The treatments with 2 g L^−1^ (63.28 mg) and 1 g L^−1^ of chitosan (58.33 mg) also significantly improved root dry biomass with respect to the carbendazim treatment (56.35 mg). All treatments were superior (*p* < 0.05) to the treatment with *F. solani* alone (30.48 mg).

In terms of root length, the 3 g L^−1^ chitosan treatment (49.25 mm) showed significantly better results (*p* < 0.05) than the 1 and 2 g L^−1^ chitosan and synthetic fungicide treatments (38.08 mm). The chitosan treatments performed better than the synthetic fungicide. The lowest results were obtained with the *F. solani* treatment. Chitosan could protect root integrity and growth under *F. solani* stress conditions (Table 1).

Figure 4 shows the disease severity (DS%) and disease incidence (DI%) caused by *F. solani* on tomato plants treated with different concentrations of chitosan and a synthetic fungicide. Treatments with chitosan showed a significant reduction in both DS and DI compared to *F. solani*. The 3 g L^−1^ chitosan treatment presented the lowest DS and DI (18.33 and 44.44%, respectively), showing significant differences (*p* < 0.05) with *F. solani* and the synthetic fungicide (carbendazim).

The 1 and 2 g L^−1^ chitosan treatments showed no significant differences (*p* > 0.05) in terms of DS compared to the synthetic fungicide treatment. However, both treatments showed a lower DS compared to plants infested only with *F. solani*. Regarding DI, the 2 g L^−1^ treatment was also superior to the synthetic fungicide. The lowest concentration of chitosan (1 g L^−1^) did not show significant differences (*p* > 0.05) to the synthetic fungicide. The *F. solani* treatment showed the highest DS and DI (85% and 100%, respectively), highlighting the need for effective treatments for the control of *F. solani* on tomato plants.

Figure 5 shows the biocontrol efficacy of different treatments on tomato plants infested with *F. solani*. The treatment with the highest concentration of chitosan (3 g L^−1^) showed the highest biocontrol efficacy, reaching 80% and showing significant differences (*p* < 0.05) with the 1 g L^−1^ and the synthetic fungicide treatments. However, it was not significantly different (*p* > 0.05) from the 2 g L^−1^ treatment. The 1 and 2 g L^−1^ chitosan treatments showed no significant differences with the synthetic fungicide (*p* > 0.05) (Figure 5).

The SOD activity in tomato plants infested with *F. solani* varied significantly (*p* < 0.05) across treatments (Table 2). Plants treated with 3 g L^−1^ chitosan showed the highest activity (2.70 U mg^−1^ protein), showing significant differences (*p* < 0.05) with the rest of the treatments. The 2 and 3 g L^−1^ chitosan treatments were superior to the synthetic fungicide treatment. However, the 1 g L^−1^ treatment showed no difference (*p* > 0.05) with the synthetic fungicide. The control showed the lowest activity (0.86 U mg^−1^ protein) (Table 2).

Plants treated with 3 g L^−1^ chitosan showed the highest CAT activity (7.35 U mg^−1^ protein). All chitosan treatments had better results than the synthetic fungicide (carbendazim) and the *F. solani* treatment (Table 2).

Concentrations of 2 and 3 g L^−1^ of chitosan showed no significant differences (*p* > 0.05) in POX activity, but the 3 g L^−1^ treatment had better results than the rest of the treatments. The 1 and 2 g L^−1^ chitosan treatments showed no significant differences with the carbendazim treatment but did have significant differences with the other treatments (Table 2).

The 3 g L^−1^ treatment showed the highest PAL activity (4.58 min mg^−1^ protein). All chitosan treatments showed higher PAL activity with respect to the *F. solani* treatment and the control. The 1 and 2 g L^−1^ chitosan treatments showed no significant differences (*p* > 0.05) with the carbendazim treatment. The lowest results in enzyme activity (SOD, CAT, POX and PAL) were always obtained with the control treatment (no application) (Table 2).

## 3. Discussion

The results obtained demonstrate the effectiveness of chitosan in inhibiting the mycelial growth of *F. solani*. The treatment of 3 g L^−1^ of chitosan showed the best results in inhibiting mycelial growth and biomass of *F. solani*. This treatment was also better than the synthetic fungicide used as a positive control.

This effect could be attributed to the great amount of chitosan that can interact with the fungal cell walls, interfering with their growth and development. The positive charges on chitosan’s amino group bind to negatively charged substances in the fungal cell wall, such as phospholipids, forming a polymeric membrane on the fungal cell surface and altering its permeability [34]. In addition, chitosan induces the production of reactive oxygen species (ROS), causing oxidative damage to the mycelial growth of fungi and inhibiting spore germination [26].

These results are similar to those reported by Torres-Rodriguez et al. [35]. The authors showed that the highest concentration of chitosan (3 g L^−1^) was capable of the highest inhibition of *F. oxysporum* with a PMGI of 80% and no significant differences with the synthetic fungicide.

In another study, different concentrations of chitosan (0.125, 0.25, 0.5, 1 and 2 g L^−1^) were evaluated and it was found that its mycelial and biomass growth inhibitory activity upon *F. solani* increased significantly according to chitosan concentration. The results showed that all tested concentrations of chitosan exhibited antifungal activity as opposed to the control [36].

In line with our results, it has been shown that the higher the concentration of chitosan, the more effective it is against phytopathogens. Mohammed et al. [37] demonstrated that chitosan, applied at a concentration of 1%, completely inhibited the mycelial growth of *Rhizoctonia solani*, the pathogen that causes black scurf on potato. Furthermore, chitosan can interfere with fungal metabolic processes, in particular those linked to energy and lipid and sugar metabolism, reducing growth and biomass [38]. Chitosan is a viable and less toxic alternative to synthetic fungicides for the control of diseases caused by *F. solani*.

The results obtained under in vitro conditions suggest that chitosan is effective in directly inhibiting the mycelial growth of *F. solani*. These results could correlate with a decrease in disease severity under in vivo conditions, considering that chitosan not only affects *F. solani* directly but also induces plant defence responses. This could reduce the ability of the phytopathogen to colonise plant tissues and limit disease progression.

SEM analysis showed cellular damage on *F. solani* caused by chitosan, leading to cell membrane disruption and cell lysis. The deformation of *F. solani* structures indicates an effective inhibition of mycelial growth and a reduction in spore viability. Our results show that chitosan acts directly on the phytopathogen, causing severe structural damage to its cell membrane and arresting its growth.

Kim et al. [39] also used SEM to observe the effects of chitosan on *F. fujikuroi*, where significant damage to the integrity of the fungal cell wall was evident. The images showed that chitosan caused porosity and cell collapse, confirming its effectiveness as an antifungal agent. These results correlate with those obtained in our study. Research into *Sclerotinia sclerotiorum* showed that chitosan treatment caused damage to the mycelial plasma membrane, with increased protein leakage and lipid peroxidation, suggesting severe structural damage similar to that observed in this paper [40]. These results suggest that chitosan’s main mode of action is cell membrane disruption, which subsequently leads to cell leakage and cell death. In addition, chitosan prevents DNA replication and causes apoptosis in phytopathogen cells [41].

The selectivity of chitosan in the management of *F. solani* is an important aspect to consider. Previous research has shown that chitosan has a more specific action on phytopathogenic fungi due to its interaction with cell wall structures, particularly with components such as chitin and glucans [16]. Beneficial organisms that do not possess these characteristics may be less affected. However, further studies are needed to determine whether the use of chitosan could negatively impact beneficial microorganisms in the soil.

*Fusarium* species can survive in soil or crop residues after harvest for prolonged periods without the presence of hosts, making them difficult to control [42]. The results obtained in this paper indicate that chitosan treatments, especially at a concentration of 3 g L^−1^, had a significant positive impact on the growth of *F. solani*-infected tomato plants.

Massoud et al. [43] showed that chitosan application at a concentration of 4 g L^−1^ reduced wilt disease in strawberries caused by *F. oxysporum*. Moreover, chitosan application increased the plants’ fresh weights, dry weights and root length compared to the control treatment. The lowest percentages of disease severity and disease incidence were also obtained with the application of chitosan, which corresponds with our study.

In this study, disease incidence was higher than disease severity. These results could be attributed to the fact that *F. solani* managed to infect the plants in a high percentage, but the level of damage caused (severity) was limited by the action of chitosan. Therefore, chitosan reduced symptom progression and prevented disease progression beyond the initial stages of infection.

In agreement with our results, the highest concentration of chitosan (4 g L^−1^) reduced the severity of potato tuber rot caused by *F. oxysporum* and *F. sambucinum* by 60 and 48.2%, respectively. Furthermore, when chitosan was applied to plants inoculated with *Fusarium* species, the plants showed a reduction in wilt severity by 33.5–45.3% compared to the control (without chitosan) [44].

The application of chitosan, this time on tomato fruits, also reduced the severity and incidence of fruit rot disease caused by *F. oxysporum*. Fruits inoculated with the highest concentration of chitosan (3 g L^−1^) showed no symptoms of the disease, results similar to those obtained in this paper [35]. DeGenring et al. [45] showed that pre-harvest applications of chitosan reduced the incidence and severity of apple scab by up to 55% compared to the control (water only).

These results may be due to chitosan’s ability to induce defence responses in plants. Chitosan activates the synthesis of phytoalexins and defence enzymes and the increased production of callose cell wall appositions in the epidermis and outer cortex of the host, which strengthens plant resistance against phytopathogens such as *Fusarium* spp. [16,22].

Plants mobilise a great deal of energy to activate their defence mechanisms against phytopathogens. This includes not only the direct fight against infection, such as water flow management and CO₂ assimilation, but also the management of oxidative stress and the protection of the photosynthetic machinery [46]. This metabolic cost leads to a reduction in biomass [47]. In our study, chitosan treatment not only reduced the severity of *F. solani* infection but also promoted plant growth, suggesting that chitosan can mitigate the negative effects of the phytopathogen while minimising the metabolic cost associated with the activation of plant defences.

Chitosan activates plant defence responses during host–phytopathogen interactions. Chitosan treatments increased SOD, CAT, POX and PAL activity compared to synthetic fungicide-treated plants and the untreated control. These enzymes are important at different stages of the disease cycle. For example, SOD and CAT neutralise reactive oxygen species generated in the early stages of plant-pathogen interaction, protecting plant cells from oxidative damage. POD and PAL contribute to cell wall fortification through lignification and the synthesis of phenolic compounds, which reinforces the plant’s physical and chemical barriers at later stages of the disease cycle. These combined mechanisms may explain the observed reduction in disease severity following chitosan application.

These results show a dual action of chitosan causing physical damage to the cell membrane of plant pathogens and activating defence enzymes in plants. This shows that root rot management involves both inhibiting the growth of the phytopathogen and strengthening the plant’s defensive capacity.

Our results are in line with recent studies on potato plants in which chitosan increased the expression of defence-related genes and the synthesis of defence-related enzymes such as peroxidase and polyphenoloxidase [37].

SOD is an important enzyme in the removal of ROS, which accumulate in response to biotic stress. The increased SOD activity in chitosan-treated plants indicates an efficient reduction of oxidative damage caused by *F. solani* infection, protecting plant cells from oxidative stress [48,49].

Catalase decomposes hydrogen peroxide (an ROS) into water and oxygen, minimising cell damage. Therefore, chitosan not only reduces oxidative stress by increasing SOD activity, it also efficiently removes hydrogen peroxide, another toxic by-product of oxidative stress [48,50].

POX are essential in the lignification and suberisation of cell walls, strengthening physical barriers against invasion by phytopathogens [51,52]. The high POX activity suggests that chitosan enhances the plant’s ability to reinforce its structural defences.

PAL is a key enzyme in the phenylpropanoid pathway, responsible for the synthesis of phenolic compounds, such as phytoalexins, lignin and other anti-microbial compounds. These compounds enhance plants’ defences against phytopathogens and strengthen cell walls, which improves their structural and chemical resistance against infections [53,54,55]. The increased activity of PAL indicates an increased production of these defensive compounds in response to chitosan treatment, which contributes to plants’ resistance to *F. solani*.

Chitosan has been shown to increase the activity of SOD, POX and CAT, i.e., enzymes involved in the direct neutralisation of ROS [56]. A study on *Arabidopsis* seedlings treated with chitosan revealed that chitosan application increased the activity of POX and other defence-related enzymes, confirming their role in inducing defensive responses in plants against phytopathogens [57]. These results suggest that chitosan, especially at the concentration of 3 g L^−1^, is effective for the biocontrol of *F. solani* on tomato plants, as well as being able to reduce the use of traditional synthetic fungicides.

## 4. Materials and Methods

### 4.1. Fusarium solani

The fungus, originally isolated from tomato plants affected by *F. solani* [58], was subsequently donated to the Plant Pathology Laboratory of the Centro de Investigaciones Biológicas del Noroeste (CIBNOR) in La Paz, Baja California Sur, Mexico.

### 4.2. Chitosan’s Antifungal Activity Upon Fusarium solani Mycelial Growth

First, 5 mm discs of *F. solani* were placed in the centre of Potato Dextrose Agar (PDA) Petri dishes containing different treatments of low molecular weight (160 kDa, 85% DD) chitosan (1, 2 and 3 g L^−1^) according to the modified methodology of Tikhonov et al. [59]. As a control, a set of plates was inoculated with the phytopathogen and treated with the synthetic fungicide carbendazim (6 mg mL^−1^) and another set was inoculated only with *F. solani*. Plates were incubated at 28 °C for 7 days. Mycelial growth was measured using a digital vernier when the mycelium reached the edges of the control plates and the result was expressed as the average diameter. The PMGI (%) of *F. solani* was calculated using the formulaPMGI (%) = [(R1 − R2)/R1] × 100(1)
where R1 represents mycelial growth on the control plates and R2 represents mycelial growth on the chitosan-treated plates. Six replicates (six petri dishes) per treatment were performed.

### 4.3. Chitosan’s Antifungal Activity Upon Fusarium solani Biomass

A suspension of *F. solani* spores (1 × 10^6^ conidia mL^−1^) was added to Potato Dextrose Broth (PDB) liquid culture medium with different chitosan concentrations (1, 2 and 3 g L^−1^) according to the methodology of Torres-Rodriguez et al. [35]. The mixtures were incubated for 7 days at 28 °C with constant shaking at 170 r/min. Flasks containing *F. solani* plus carbendazim (6 mg mL^−1^) and flasks containing only *F. solani* in PDB were used as controls. After the incubation period, mycelia were filtered through a 30 μm-pore-sized mesh, washed with distilled water and collected by centrifugation at 5000 rpm; this was repeated twice. The collected mycelia were dried in an oven and weighed to determine the biomass of *F. solani*. Six replicates per treatment were performed.

### 4.4. The Effect of Chitosan on Fusarium solani Cell Structure

From the treatments of the in vitro antifungal effect experiment, a thin surface layer of *F. solani* agar was cut to determine the damage to the cell structure by Scanning Electron Microscopy (SEM) according to the modified methodology of Brilhante et al. [60]. The samples were placed on a coverslip attached to an aluminium stub. They were placed in a chamber saturated with OsO_4_ (osmium tetroxide) for 72 h. Dehydration continued in a chamber with granulated silica and filter paper for one week. Once the samples were completely dehydrated, they were coated in gold using a Sputter Coater (Denton Vacuum Model, Desk II). Once coated, they were observed with a Scanning Electron Microscope (S-3000N Model, Hitachi, Tokyo, Japan) and a Quartz PCI image processor.

### 4.5. Biocontrol of Chitosan Against Fusarium solani in Tomato

#### 4.5.1. Treatments and Growth Conditions

*F. solani* was grown on PDA at 28 °C for a period of 7 days. Then, 5 mL of sterile water was added and spores were scraped off and transferred to 250 mL Erlenmeyer flasks with 100 mL of PDB. They were incubated at 150 rpm for 5 days, adjusting the inoculum to a final concentration of 1 × 10^6^ conidia mL^−1^.

Saladette tomato plants were grown in 200-cavity germination trays using COSMOPEAT as substrate. They were maintained at 25 °C, with a relative humidity of 80% and a photoperiod of 12 h of light, in a Conviron GEN2000 growth chamber (Controlled Environments Limited, Winnipeg, MB, Canada). After 25 days of growth, one-third of the plant roots were trimmed and dipped in the *F. solani* conidial suspension for 15 min before transplanting. One group of plants was inoculated only with distilled water as a control. Six replicates per treatment (six tomato plants) were performed in a completely randomised experimental design.

Twenty-four hours before inoculation with *F. solani*, the following treatments were applied: [T1] 1 g L^−1^ Chitosan + *F. solani*, [T2] 2 g L^−1^ Chitosan + *F. solani*, [T3] 3 g L^−1^ Chitosan + *F. solani*, [T4] Synthetic fungicide (carbendazim) + *F. solani*, [T5] *F. solani* and [T6] Control (distilled water). Chitosan was applied near the root collar and leaves with an atomiser until the leaves were completely covered. Plants were maintained at 28 °C, 80% RH and 12 h light in a Conviron growth chamber for 28 days.

#### 4.5.2. Impact of Chitosan on the Growth of Tomato Plants Infested with Fusarium solani

The effect of chitosan on different growth variables of tomato plants infested with *F. solani* was evaluated. The variables analysed were plant height, stem diameter, dry biomass and root length, which were measured 28 days after applying the treatments [10]. Additionally, leaf samples were collected and stored at −80 °C for future analyses of enzyme activity.

#### 4.5.3. Assessment of Disease Severity

Disease severity (DS) was assessed 28 days after the application of the treatments using the scale developed by Marlatt et al. [61]. The scale used was as follows: 0 = plants without symptoms; 1 = slight chlorosis; 2 = slight chlorosis and wilting or stunting; 3 = moderate chlorosis and wilting or stunting; 4 = severe chlorosis and wilting or stunting; and 5 = dead plants. Subsequently, a DS index was estimated using the following formula:(2)$$DS(\%)=\left[\sum_{i=0}^{5}ni\left(sti\right)/\left(N\times K\right)\right]\times 100$$
where ni = number of plants at DS stage of development, sti = DS stage (0–5), N = total number of plants assessed and K = highest possible score (5).

#### 4.5.4. Assessment of Disease Incidence

Disease incidence with chitosan application (DI) was determined as the number of plants showing disease symptoms, such as chlorosis and wilting or stunting, divided by the total number of plants [62]. The percentage of disease incidence was determined using the following formula:DI (%) = (Pi/TP) × 100 (3)
where Pi is the number of infected plants and TP is the total number of plants.

The experiment consisted of six replicates per treatment and was repeated twice.

Treatment efficiency: The efficiency of chitosan application against the phytopathogen was evaluated using the following formula proposed by Abbott [63]:E = [(FIWoQ − FIWQ)/FIWoQ] × 100 (4)
where E = Efficiency (%); FIWoQ = Disease severity in the control (without chitosan); FIWQ= Disease severity with chitosan application.

#### 4.5.5. Antioxidant Enzyme Activity in Tomato Plants

Leaf samples frozen at −80 °C from the treatments described above were homogenised with glass beads and mechanical agitation to disintegrate the tissue with a homogeniser (MPI, Fast Prep-24, Irvine, CA, USA). To each Eppendorf tube, 1 mL of phosphate buffer (100 mM, pH 7.0) was added. Samples were centrifuged at 3800 rpm for 20 min at 4 °C and the supernatant was subjected to enzyme assays.

##### Catalase

CAT enzyme activity was determined by mixing 100 μL of 100 mM phosphate buffer with 30 μL of methanol, 20 μL of the sample extract and 20 μL of H_2_O_2_ (35.2 µM). The plate was immediately covered and incubated and shaken at room temperature for 20 min. Then, 30 μL of KOH (10 M) and 30 μL of purpald reagent (23.5 M) at 1% in 5% HCl solution were added. This was incubated for a further 10 min. Finally, 10 μL of 0.5 M potassium periodate in 0.5 M KOH solution was added. Spectrophotometer readings were taken at 540 nm. One unit of CAT activity is defined as the amount of enzyme that reacts with 1 nmol of formaldehyde per minute and is expressed as U mg^−1^ of protein [64].

##### Peroxidase

POX enzyme activity was determined by mixing 300 μL of phosphate buffer (50 mM), 5 μL of guaiacol solution (3.33 mM), 10 μL of the sample extract and 3 μL of an H_2_O_2_ solution (4 mM). After mixing, it was incubated at 30 °C for 10 min. The spectrophotometer reading was at 470 nm. The unit of POX activity was defined as the amount of enzyme causing tetraguaiacol formation in the presence of H_2_O_2_ per min and was expressed in U mg^−1^ of protein [65].

##### Superoxide Dismutase

SOD enzyme activity was determined by mixing 1200 μL of phosphate solution (0.1 M), 20 μL of sample extract and 20 μL of xanthine oxidase solution (0.1 U of xanthine oxidase in 1 mL of 2 M ammonium sulphate). The absorbance change at 560 nm was recorded every 20 s for 5 min. One unit of SOD activity was defined as the amount of enzyme required to inhibit 50% of the O_2_ reaction in the presence of nitro-blue tetrazolium (NBT) reagent and was expressed as U mg-1 protein [66].

##### Phenylalanine Ammonia-Lyase

To quantify PAL activity, 150 μL of sample extract, 200 μL of phenylalanine (40 mM) and 450 μL of Tris HCl buffer (0.1 M) were mixed together. This was incubated at 37 °C for 30 min. The reaction was stopped with 200 μL of 25% trichloroacetic acid (TCA). The mixture was centrifuged at 10,000 rpm for 15 min at 4 °C. PAL activity was determined in relation to the cinnamic acid formed at 290 nm absorbance and the prepared standard curves. One unit of PAL activity is defined as 1 mmol of cinnamic acid formed per minute per milligram of protein (min mg^−1^ protein) [67].

### 4.6. Statistical Analysis

Data were analysed by one-way analysis of variance (ANOVA) using STATISTICA 10.0 software (StatSoft software, Tulsa, OK, USA) and Tukey’s test (*p* < 0.05) was used to separate the means. Data were tested for normality using the Shapiro-Wilk test and homogeneity of variances using Bartlett’s test before the ANOVA.

## 5. Conclusions

Chitosan is effective in inhibiting the mycelial growth and biomass of *F. solani*. The 3 g L^−1^ treatment was the most effective, outperforming even the synthetic fungicide carbendazim. SEM revealed that chitosan causes significant damage to the cell structure of *F. solani*, including cell lysis and deformation of mycelium and spores. This severe structural damage reinforces the efficacy of chitosan as an antifungal agent, as it interferes with the cellular integrity of the *F. solani*, inhibiting its growth.

Treatment with chitosan not only reduced the disease severity and incidence caused by *F. solani* but also caused positive effects on the growth of tomato plants. Plants treated with 3 g L^−1^ chitosan had increased height, stem diameter, root dry biomass and root length compared to plants from the synthetic fungicide and *F. solani* (no chitosan) treatments.

Chitosan treatments significantly increased the activity of defence enzymes such as SOD, CAT, POX and PAL. The SOD, CAT and POX are important in the neutralisation of ROS and PAL is involved in the fortification of cell walls, improving plant resistance to *F. solani*. Given its natural origin and low toxicity, chitosan represents a promising option for integrated disease management in agriculture, contributing to more environmentally friendly and sustainable agricultural practices.

## Figures and Tables

**Figure 1 plants-14-00431-f001:**
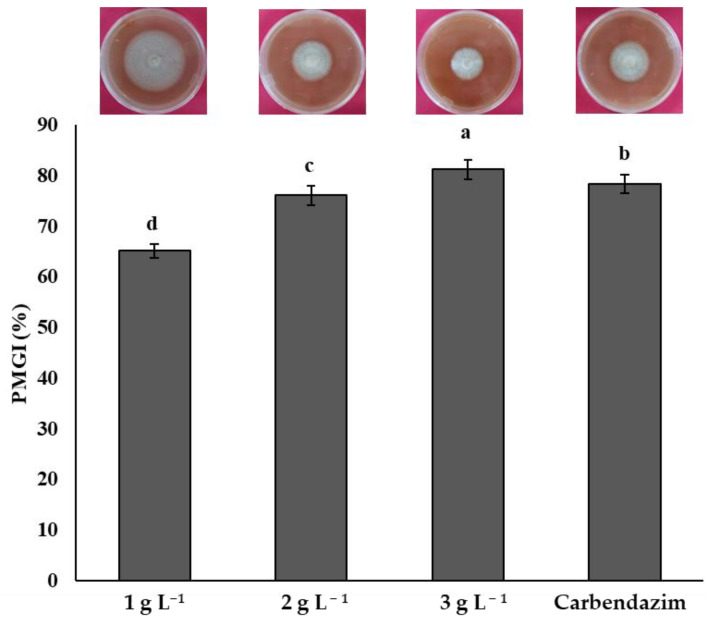
Percentage of mycelial growth inhibition (PMGI) of *Fusarium solani* by chitosan. Means with the same letters do not differ significantly (*p* < 0.05) according to Tukey’s test. Data are presented as the mean ± standard deviation of six replicates (six Petri dishes per treatment).

**Figure 2 plants-14-00431-f002:**
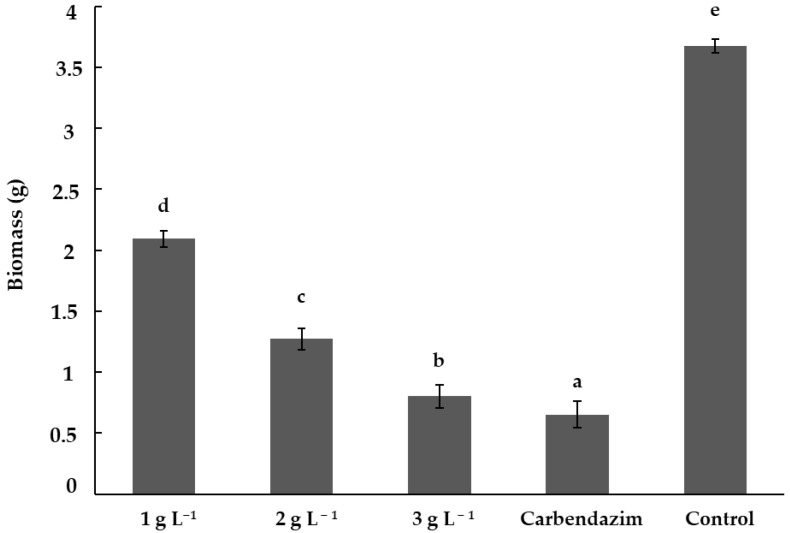
Inhibition of *Fusarium solani* biomass by chitosan. Means with the same letters do not differ significantly (*p* < 0.05) according to Tukey’s test. Data are presented as the mean ± standard deviation of six replicates (six Erlenmeyer flasks per treatment).

**Figure 3 plants-14-00431-f003:**
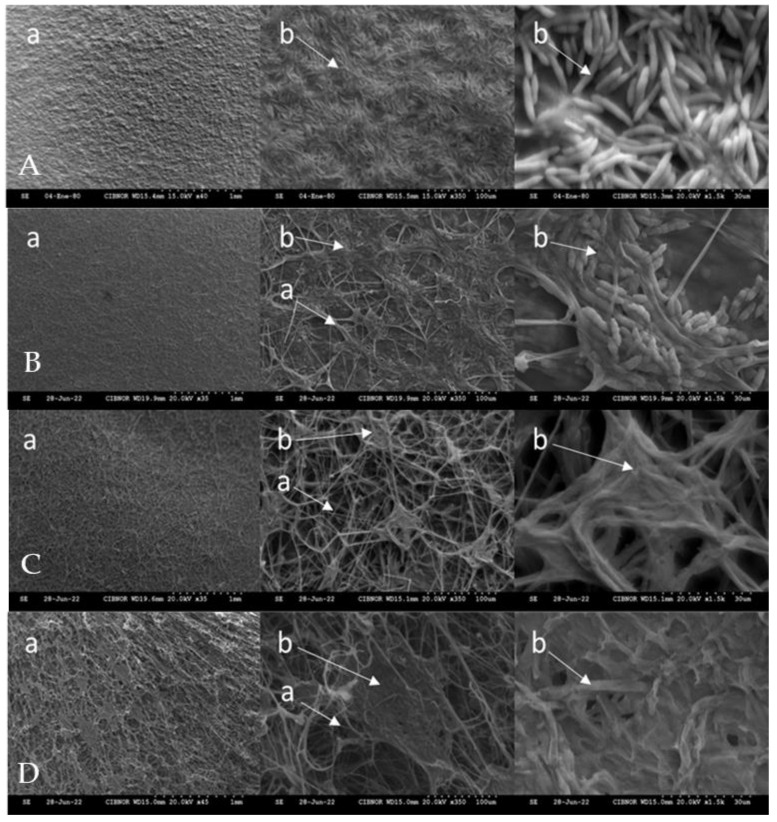
Effect of chitosan on the cell structure of *Fusarium solani*. **A**: control treatment, **B**: treatment 1 g L^−1^, **C**: treatment 2 g L^−1^, **D**: treatment 3 g L^−1^, a: mycelium of *Fusarium solani*; b: spores of *Fusarium solani*.

**Figure 4 plants-14-00431-f004:**
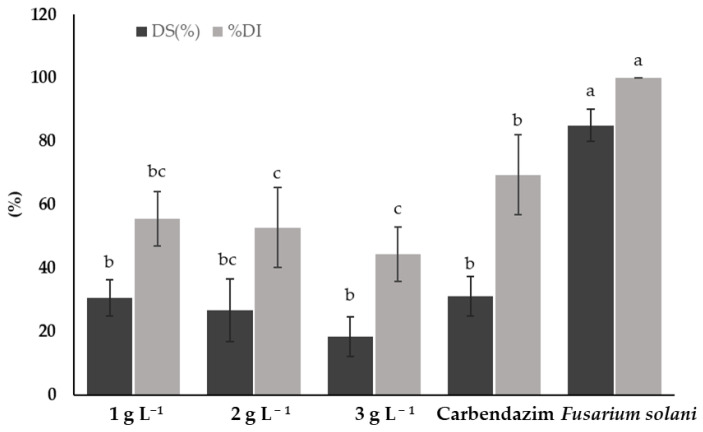
Disease severity (DS) and disease incidence (DI) caused by *Fusarium solani* on tomato plants treated with chitosan. Means with the same letters do not differ significantly (*p* < 0.05) according to Tukey’s test. Data are presented as the mean ± standard deviation of six replicates (six tomato plants per treatment).

**Figure 5 plants-14-00431-f005:**
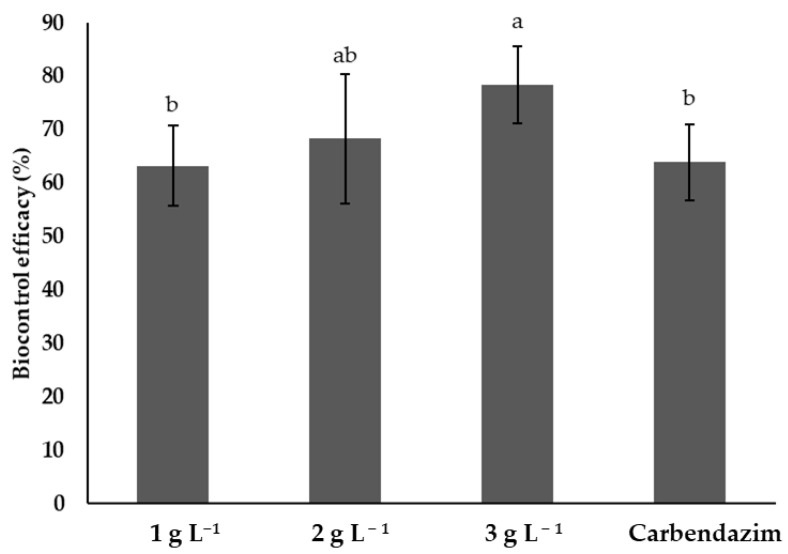
Efficiency of different treatments in the control of *Fusarium solani* on tomato plants. Means with the same letters do not differ significantly (*p* < 0.05) according to Tukey’s test. Data are presented as the mean ± standard deviation of six replicates (six tomato plants per treatment).

**Table 1 plants-14-00431-t001:** Impact of chitosan on the growth of tomato plants infested with *Fusarium solani*.

Treatment	Plant Height (mm)		Stem Diameter (mm)		Dry Root Biomass (mg)		Root Length (mm)	
1 g L*^−^*^1^ Ch + F	293.73 ^d^	±0.57	2.95 ^bc^	±0.19	58.33 ^c^	±0.49	39.50 ^d^	±0.53
2 g L^−1^ Ch + F	321.67 ^bc^	±1.45	3.7 ^ab^	±0.24	63.28 ^b^	±0.29	44.20 ^c^	±0.47
3 g L*^−^*^1^ Ch + F	330.42 ^b^	±1.29	4.29 ^a^	±0.51	68.33 ^a^	±0.67	49.25 ^b^	±0.70
Carbendazim	309.82 ^cd^	±0.92	3.62 ^ab^	±0.47	56.35 ^d^	±0.82	38.08 ^e^	±0.53
*Fusarium solani*	155.90 ^e^	±1.07	2.32 ^c^	±0.61	30.48 ^e^	±0.72	16.25 ^f^	±0.34
Control	350.82 ^a^	±1.05	4.31 ^a^	±0.89	69.33 ^a^	±0.90	52.25 ^a^	±0.77

Ch + F: chitosan + *Fusarium solani*, Control: distilled water. Means with the same letters do not differ significantly (*p* < 0.05) according to Tukey’s test ± standard deviation.

**Table 2 plants-14-00431-t002:** Enzyme activities in tomato plants infested with *Fusarium solani* and treated with chitosan.

Treatment	SOD Activity (U mg^−1^ Protein)	CAT Activity (U mg^−1^ Protein)	POX Activity (U mg^−1^ Protein)	PAL Activity (min mg^−1^ Protein)
1 g L*^−^*^1^ Ch + F	2.08 ± 0.29 ^bc^	6.17 ± 0.22 ^c^	5.83 ± 0.26 ^b^	3.42 ± 0.39 ^b^
2 g L^−1^ Ch + F	2.18 ± 0.32 ^b^	6.70 ± 0.26 ^b^	6.45 ± 0.27 ^ab^	3.70 ± 0.54 ^b^
3 g L^−1^ Ch + F	2.70 ± 0.26 ^a^	7.35 ± 0.29 ^a^	6.78 ± 0.20 ^a^	4.58 ± 0.41 ^a^
Carbendazim	1.61 ± 0.29 ^cd^	4.93 ± 0.19 ^d^	6.00 ± 0.70 ^b^	3.30 ± 0.43 ^b^
*Fusarium solani*	1.51 ± 0.28 ^d^	3.77 ± 0.22 ^e^	3.81 ± 0.35 ^c^	1.88 ± 0.30 ^c^
Control	0.86 ± 0.23 ^e^	1.27 ± 0.11 ^f^	1.75 ± 0.38 ^d^	1.08 ± 0.30 ^d^

Ch + F: chitosan + *Fusarium solani*, Control: distilled water. Means with the same letters do not differ significantly (*p* < 0.05) according to Tukey’s test ± standard deviation.

## Data Availability

The raw data supporting the conclusions of this article will be made available by the authors on request.

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
