# Peer review of "Biocontrol of Fusarium solani: Antifungal Activity of Chitosan and Induction of Defence Enzymes"

_plants, 2025, doi:10.3390/plants14030431_

Round 1

Reviewer 1 Report

Comments and Suggestions for Authors

Dear authors

The manuscript “Biocontrol of Fusarium solani: Antifungal Activity of Chitosan and Induction of Defence Enzymes” aimed to prove the efficiency of chitosan against Fusarium solani on tomato plants, quantifying its antifungal and resistance induction activity. The results showed that all chitosan treatments significantly inhibited the mycelial growth and biomass of F. solani. Scanning electron microscopy (SEM) revealed that chitosan causes severe structural damage to F. solani. Enzyme assays showed that chitosan significantly increased superoxide dismutase (SOD), catalase (CAT), peroxidase (POD) and phenylalanine ammonia-lyase (PAL) activity, indicating an increased defensive response. This manuscript is very interesting and advances the clarification of the mechanisms involved in the control of tomato fusariosis. I recommend its publication, as long as a thorough review is carried out to consider the points indicated below.

1)    Please make clear in the text the name of the disease caused in tomato plants by the pathogen Fusarium solani. It is necessary to make it clear to the reader that the causal agent is Fusarium solani and that the disease involves the interaction between pathogen and host. It has not been clearly described which mechanisms are involved in controlling the pathogen and which are involved in controlling the disease.

2)    Another point that must be corrected is that chitosan is not a biological agent, but a derivative of a crustacean, therefore a molecule, or a complex of molecules.

3)    In the introduction, it should be mentioned which fungicide is used to control Fusariosis, what is the mechanism? Would be similar the same one be used as a control in the manuscript?

4)    What about the selectivity of chitosan? Would it have the same effects on the natural enemies of F. solani?

5)    In Material and Methods, lines 312 and 325 the PDA translation is not complete.

6)    In the description of the tests carried out in vitro, the experimental design and the number of plants (n) that make up a plot were not described.

7)    Regarding the growth promotion assay, I think the interpretation should be based on a control containing only the chitosan treatment, without the pathogen. The interpretation of the test as it stands must be limited to highlighting that plants challenged with F.solani and treated with chitosan (3g.L) presented the same growth measurements as the positive control.

8)    It cannot be said that there is a correlation between doses and control efficiency without carrying out a regression analysis, or even a correlation between the parameters evaluated.

9)    And regarding the data presented on doses, it would be important and correct to present a regression analyses to define the best dose, since 4 (0, 1, 2, 3 gL) were used.

10) Reformat the figure containing the SEM images, removing the dose of chitosan from the middle of the figure. Transfer this information to the legend. Were there no photos taken with the carbendazin treatment?

11) In figures 1 and 2, check whether the statistical differences are correct. According to the SD presented in the bars, it seems to me that there is no difference between treatment with 3 gL and carbendazin. Please add the number of plants that make up these averages.

12) In the discussion, the authors need to reduce the comparison with others authors and discuss the interaction with the results obtained. For example, what is the relationship between data obtained in vitro and suppression of severity? Why was the incidence greater than the severity? How and at what point in the disease cycle would the quantified enzymes be acting?

13) And what is the efficiency of resistance induction for a nercrotrophic pathogen. Is Fusarium solani biotrophic? nercrotrophic? hemibiotrophic?

Author Response

Dear Reviewer 1.
Thank you very much for each of your comments, which have undoubtedly been valuable for the improvement of our manuscript.
Please find attached the response to each of your comments.

Reviewer 2 Report

Comments and Suggestions for Authors

The article investigates the efficacy of chitosan as a biocontrol agent against F. solani, a major pathogen affecting tomato plants. The study provides valuable insights into the mechanisms by which chitosan enhances the plant's defense responses and mitigates the negative effects of the pathogen. The authors demonstrate that chitosan treatment not only reduces disease severity but also promotes plant growth, suggesting its dual benefits in managing the pathogen and improving plant performance. Most part of the article is well-written except the discussion section

My minor concerns include the size of columns in Fig. 1, 2, 4 and 5. The size of the columns can be reduced. Fig. 1 and 2 can be combined together, while Fig. 4 and 5 can be combined together. In addition, the Discussion should be written in a more concise manner and better follow the academic writing conventions.

Author Response

Dear Reviewer 2.
Thank you very much for each of your comments, which have undoubtedly been valuable for the improvement of our manuscript.
Please find attached the response to each of your comments.

Round 2

Reviewer 1 Report

Comments and Suggestions for Authors

Dear authors,

All my considerations were met, with the exception of the regression analysis, which was justified by the author.

Author Response

Dear Reviewer 1.

We appreciate the suggestion to perform a regression analysis to determine the best chitosan dose. However, the main objective of our study was to compare the effect of different discrete concentrations of chitosan on the inhibition of Fusarium solani and the induction of defensive responses in tomato plants. For this purpose, we used a multiple comparison analysis (Tukey's test), which allowed us to statistically identify the most effective treatment among the doses evaluated. We believe that a regression analysis could provide additional information on the dose-response relationship. However, given that the treatments selected were discrete categories and no experimental design with smaller or continuous dose intervals was considered, we believe that the approach used is adequate to meet the objectives set out in this study.

Best regards